# Self-Matrix N-Doped Room Temperature Phosphorescent Carbon Dots Triggered by Visible and Ultraviolet Light Dual Modes

**DOI:** 10.3390/nano12132210

**Published:** 2022-06-28

**Authors:** Huiyong Wang, Hongmei Yu, Ayman AL-Zubi, Xiuhui Zhu, Guochao Nie, Shaoyan Wang, Wei Chen

**Affiliations:** 1School of Chemical Engineering, University of Science and Technology Liaoning, Anshan 114051, China; hywang@163.com (H.W.); aswsy64@163.com (S.W.); 2Department of Physics, The University of Texas at Arlington, Arlington, TX 76019-0059, USA; ayman.alzubi@uta.edu; 3Department of Chemical Engineering, Yingkou Institute of Technology, Yingkou 115014, China; 4School of Physics and Telecommunication Engineering, Yulin Normal University, Yulin 537006, China; bccu518@ylu.edu.cn

**Keywords:** carbon dots, room temperature phosphorescence, visible/ultraviolet light excitation, anti-counterfeiting

## Abstract

The synthesis of room temperature phosphorescent carbon dots (RTP-CDs) without any matrix is important in various applications. In particular, RTP-CDs with dual modes of excitation are more interesting. Here, we successfully synthesized matrix-free carbonized polymer dots (CPDs) that can generate green RTP under visible and ultraviolet light dual-mode excitation. Using acrylic acid (AA) and ammonium oxalate as precursors, a simple one-pot hydrothermal method was selected to prepare AA-CPDs. Here, acrylic acid is easy to polymerize under high temperature and high pressure, which makes AA-CPDs form a dense cross-linked internal structure. Ammonium oxalate as a nitrogen source can form amino groups during the reaction, which reacts with a large number of pendant carboxyl groups on the polymer chains to further form a cross-linked structure. The carboxyl and amino groups on the surface of AA-CPDs are connected by intermolecular hydrogen bonds. These hydrogen bonds can provide space protection (isolation of oxygen) around the AA-CPDs phosphor, which can stably excite the triplet state. This self-matrix structure effectively inhibits the non-radiative transition by blocking the intramolecular motion of CPDs. Under the excitation of WLED and 365 nm ultraviolet light, AA-CPDs exhibit the phosphorescence emission at 464 nm and 476 nm, respectively. The naked-eye observation exceeds 5 s and 10 s, respectively, and the average lifetime at 365 nm excitation wavelength is as long as 412.03 ms. In addition, it successfully proved the potential application of AA-CPDs in image anti-counterfeiting.

## 1. Introduction

Room temperature phosphorescence (RTP) has a longer life than fluorescence and has a wide range of applications in safety [1,2], optoelectronic devices [3,4], and biological imaging [5,6]. In fact, phosphorescence is difficult to achieve at room temperature due to spin prohibition, the probability of transitions between singlet (S_1_) and triplet states (T_1_) is extremely low, and triplet phosphorescence is easily quenched by oxygen [7,8]. However, recent studies have found that the possibility of adjusting the properties of afterglow emission through simple structural modifications [9,10,11,12,13] looks promising. CDs, as an emerging carbon-based luminescent nanomaterial, has attracted more and more attention for its RTP performance due to its low cost, convenient preparation, good stability, environmental friendliness, and low toxicity [14,15,16]. At present, the RTP phenomenon based on CDs is more through embedding in various substrates, including polyvinyl alcohol [13], polyacrylamide **[17]**, polyurethane [18], urea/biuret [10], boric acid [19], layered double hydroxides [20], etc. Although the introduction of substrates can achieve the RTP of CDs, the inherent chemical and physical properties of substrates hinder the RTP properties and applications of CDs [21].

In recent years, matrix-free RTP CDs materials were obtained through hydrogen bonds formed by the internal functional groups of CDs, which can effectively inhibit intramolecular vibration and rotation [2,22,23]. Meanwhile, the complex preparation process, high toxicity, high cost, and potential environmental hazards caused by the method of embedding matrix are also solved [24]. Specifically, the introduction of -COOH and -OH groups facilitates the formation of hydrogen bonds, which can provide spatial protection around the phosphor and further stabilize triplet excitons of the RTP emission [25]. The doping of N, P, or halogen in CDs promotes the generation of n-π* transition, which promotes the transformation of excitons into triplet states through intersystem crossover (ISC) [1,2,22,23]. Moreover, the doped CDs can reduce energy gap (ΔE_ST_) [1], which is conducive to the self-fixation of triplet excitons.

However, the vast majority of reported matrix-free RTP-CDs materials need to be excited under ultraviolet light, which greatly limits their application (such as biology related fields [26]). Compared with ultraviolet light, visible light is less phototoxic, more penetrating, and more likely to trigger RTP [21,27,28]. Hu [21] et al. reported that orange afterglow of CDs prepared from L-aspartic acid as raw material can be observed under commercial blue LED (420 nm), suggesting that L-aspartic acid could form a structure similar to crosslinked polymer at high temperature. In fact, the CDs obtained due to the incomplete carbonization of the polymer clusters should be classified as carbonized polymer dots (CPDs) [29]. CPDs exhibit a polymer/carbon hybrid structure, which not only have excellent optical properties, but also inherit the properties of polymers and special photoluminescence (PL) mechanism [30]. The CPDs formed by polymerization of certain structures may produce RTP emissions. This process is considered to be able to self-fix to excite the triplet state to form a more compact core structure, which is similar to the process of embedding in a solid matrix [31]. Of course, effective ISC is another key factor that must be considered to implement RTP.

In this work, we found that the CDs prepared with acrylic acid (AA) as carbon source and ammonium oxalate as nitrogen source were CPDs with self-matrix properties, namely AA-CPDs, which can be used as a solidified host or as a luminescent guest without any matrix doping. It has the property of generating green RTP emission under the dual-mode excitation of visible/ultraviolet light. This choice is based on the following considerations: (i) Acrylic acid can be polymerized and forms polymer at high temperature. The high-temperature and high-pressure conditions of the hydrothermal process increase the collision, entanglement, and crosslinking between polymer chains. Therefore, AA-CPDs have a tighter and better cross-linked internal structure. (ii) The -OH groups and the O atoms of C=O can be used as multiple reaction sites for cross-linking polymerization to form intramolecular hydrogen bonds with amino groups, respectively. Amino groups have been proven to be an effective sub-fluorophore for fluorescence emission [32]. (iii) The N atoms of amino groups are conducive to the n-π* transition, thereby promoting the effective filling of triplet excitons in ISC [33,34]. What is exciting is that solid AA-CPDs show a green afterglow after the visible/ultraviolet light is turned off. The afterglow is visible to the naked eye for more than 5 s and 10 s, respectively. The phosphorescence lifetime can reach 412.03 ms under 365 nm excitation, and it has stable phosphorescence performance. More importantly, AA-CPDs excited in the visible/ultraviolet light dual modes have great application potential in advanced anti-counterfeiting and hiding of complex patterns. This study provides a simple and rapid new method for preparing matrix-free RTP-CDs under the dual-mode excitation of visible/ultraviolet light.

## 2. Experimental

### 2.1. Chemicals

Acrylic acid (AA), ammonium oxalate, and oxalic acid were purchased from Aladdin (Shanghai, China). All reagents were of analytical grade and used as received without further purification. Ultrapure water was prepared by a Milli-Q ultrapure water system for all experiments.

### 2.2. Synthesis of AA-CPDs

Ammonium oxalate (0.2 g) and acrylic acid (3 mL) were dissolved in ultrapure water (12 mL). Then, the solution was subsequently transferred to a 50 mL poly(tetrafluoroethylene)-lined stainless steel autoclave, heated at 180 °C for 8 h, and cooled to room temperature. Thus, a dark-yellow suspension was obtained. The suspension was sonicated (at 40 kHz) for 10 min, filtered through a 0.22 μm filter membrane and centrifuged (at 10,000 r/min) for 15 min to remove large particles. The obtained solution was placed in a dialysis bag (Mw = 500 Da) to remove the unreacted reagents. The water in the dialysis bag should be changed frequently at the beginning of dialysis, and then changed every 4 h until it is clear and transparent for 72 h. The solution dialyzed was freeze-dried for 36 h to obtain AA-CPDs powders, then the powders were collected, sealed, and preserved for further usage.

### 2.3. Apparatus and Characterization

The morphology of AA-CPDs was measured by a JEM-ARM200F high-resolution transmission electron microscope (HRTEM) at an acceleration voltage of 200 kV (Hitachi, Tokyo, Japan). An X’Pert pro X-ray diffractometer (PANalytical, Amsterdam, The Netherlands) was used to determine the crystalline pattern of AA-CPDs. A Hyperion Fourier transform infrared (FTIR) spectrometer (Bruker, Leipzig, Germany) was used to record the FTIR spectrum of AA-CPDs. An ESCALAB 250Xi electronic spectrometer (Thermo, Waltham, MA, USA) was used to examine the X-ray photoelectron spectroscopy (XPS) spectra of AA-CPDs. A U-3900 UV–vis spectrophotometer (Hitachi, Japan) was used to measure the UV–vis absorption spectra of AA-CPDs. An FS5 spectrofluorometer (Edinburgh, England) was used to gain fluorescence and phosphorescence spectra of AA-CPDs. An Fluorolog 3-11 spectrofluorometer (Horiba, Japan) was used to determine the fluorescence lifetime of AA-CPDs. An FLS-1000 fluorescence spectrometer (Edinburgh, UK) was used to measure the phosphorescence lifetime of AA-CPDs.

## 3. Results and Discussion

### 3.1. Characterization of AA-CPDs

AA-CPDs with green phosphorescence under visible/ultraviolet light dual-mode excitation were prepared by one-step hydrothermal method using acrylic acid (AA) and ammonium oxalate as precursors without adding any acid, base, or metal ions (Figure 1a). The AA-CPDs powders show light yellow under sunlight and green phosphorescence under visible/ultraviolet dual-mode excitation (Figure 1b).

To understand the properties of the material, the morphology and particle size of the AA-CPDs were investigated and shown in Figure 2a,b. The high-resolution transmission electron microscopy (HRTEM) image (Figure 2a) of AA-CPDs reveals that AA-CPDs are nearly spherical and well-dispersed and the lattice spacing is 0.21 nm, which matches the in-plane lattice spacing of graphene (100 facet) [35]. Figure 2b shows the particle size distribution of AA-CPDs, with an average particle size of 4.7 ± 0.9 nm. It can be observed from the X-ray diffraction (XRD) pattern (Figure 2c) that the peak of AA-CPDs is located at 2θ = 20.14°, corresponding to the (100) crystal plane of graphite-like carbon [35]. The XRD pattern shows a wide diffraction band, which reveals the amorphous nature of AA-CPDs. FTIR analysis was performed to reveal information about the surface functional groups of AA-CPDs. The FTIR spectrum of AA-CPDs shown in Figure 2d exhibits that a strong and wide peak across the range of 3500–2500 cm^−1^ indicates the presence of polymerized carboxyl [36] functional groups on the surface of AA-CPDs. The peaks at 2850 cm^−1^ and 1760 cm^−1^ are the stretching vibrations of C-H [37] and C=O [38], respectively. In addition, the stretching vibration of C=N [39] (1440 cm^−1^) and the bending vibration of N-H [40] (787 cm^−1^) were detected in AA-CPDs, indicating that polyaromatic structures form in their skeletons. The results show that AA-CPDs have polymer/carbon hybridization structures consisting of hydroxyl, carbonyl, and imine groups. Notably, the existence of C=O/C=N functional groups has been reported to facilitate the generation of triple excitations through intersystem intersection (ISC) [2,41].

To further confirm the above FTIR analysis, the XPS spectra of the AA-CPDs were recorded in Figure 3a–d. The results show that AA-CPDs are composed of 73.03% C, 25.99% O, and 0.98% N (Figure 3a). The C1s XPS spectrum (Figure 3b) has three peaks at 284.6 eV, 285.2 eV, and 288.8 eV, which are attributed to C–C or C=C [42], –COOH [43], and C–N [44], respectively. The N1s XPS spectrum (Figure 3c) consists of two components: pyrrolic N (400.2 eV) [40] and amino N (401.6 eV) [44]. The O1s XPS spectrum (Figure 3d) can be deconvoluted into two peaks at 531.9 eV and 533.3 eV, indicating the existence of C–OH and C=O [40], respectively. The wide O1s band facilitates RTP production [2,41].

### 3.2. Fluorescence of AA-CPDs

The optical properties of AA-CPDs in aqueous solution and solid state were determined by UV–vis and fluorescence spectroscopy, respectively. The normalized UV–vis absorption, fluorescence excitation, and emission spectra of the AA-CPDs in aqueous solution are shown in Figure 4a. The UV–vis absorbance spectrum of the AA-CPDs (black line) showed one prominent peak centered around 210 nm. The peak was ascribed to the n–π* transition of the C=O bonds of carboxyl groups and π-π* transition of the C=C bonds comprising the aromatic network [45]. Notably, the presence of the C=O groups has previously been reported to facilitate the generation of intrinsic triplet excitons through ISC [10,41]. The emission wavelength of AA-CPDs was observed at 408 nm (blue line) under the optimal excitation wavelength of 330 nm (purple line). The excitation–emission map (Figure 4b) of the AA-CPDs in aqueous solution exhibits a single emission center (the maximum emission at 408 nm) ranging from 360 nm to 520 nm, which shows that fluorescence spectral characteristic is independent of excitation. The widespread absorption in UV regions (300–360 nm) is attributed to complex polymer/carbon hybrid structures [29,46,47]. The excitation-independent behavior of AA-CPDs in aqueous solution can be associated with the presence of one dominant fluorescence center. The best excitation peak of fluorescence excitation spectrum is 330 nm, which indicates that the C=O/C=N functional group of AA-CPDs should be responsible for fluorescence emission [21].

Further, the UV–vis absorption and fluorescence spectra of solid AA-CPDs were explored and shown in Figure 4c,d, respectively. As can be seen from Figure 4c, the UV–vis spectrum of solid AA-CPDs has two absorption peaks, including a strong absorption peak at 217 nm and a shoulder peak at 284 nm. The occurrence of acromion is due to the absorption peak formed by n-π* transition in amide functional groups. The UV–vis spectrum of solid AA-CPDs has two emission groups, which is consistent with their solid fluorescence spectrum. Fortunately, AA-CPDs powers were found to have solid fluorescence and no self-extinction [48,49]. Thanks to its inherent polymer structure, AA-CPDs successfully overcome the aggregation-induced annihilation problem faced by most solid CDs [50]. AA-CPDs, exhibit excellent solid-state fluorescence and unique RTP performance without the need for additional matrix or composite structures. The excitation and emission wavelengths of solid-state fluorescence are located at 400 nm and 478 nm, respectively. When the excitation wavelengths change from 360 nm to 440 nm, the maximum emission wavelengths of solid AA-CPDs change from 440 nm to 506 nm. It exhibits emission behavior associated with excitation wavelength, which is a common feature of CPDs [51]. The fluorescence emission wavelength of solid is red shifted compared with that of aqueous solution, which may be because the π-π* stacking interaction in solid enhances the energy transfer between AA-CPDs and generates new electronical states with emissions at long wavelength region [52,53,54]. The expansion and red shift of AA-CPDs photoluminescent peaks from aqueous solution to solid powder is attributed to the green RTP of solid powders [31].

The fluorescence decay profiles of AA-CPDs were studied at λex = 365 nm (Figure 4d). The decay curve of the AA-CPDs was fitted using a triadic exponential function with lifetimes (τ) of 0.6 ns (77.92%), 1.8 ns (21.83%) and 8.06 ns (0.25%). Average fluorescence life conforms to exponential function fit, based on Equation (1) [55]:(1)$$\tau_{avg}=\frac{\sum A_{i}\tau_{i}{}^{2}}{\sum A_{i}\tau_{i}}$$

The average fluorescence lifetime of AA-CPDs is calculated to be 1.31 ns. The absolute FL quantum yield (QY) of AA-CPDs measured under 330 nm excitation is 8.8%, showing the water-dispersed fluorescence properties.

### 3.3. Phosphorescence of AA-CPDs

The fluorescence and afterglow emission spectra of AA-CPDs at 365 nm and WLED as shown in Figure 5a. It can see that the AA-CPDs powders emit blue fluorescence at 443 nm (purple line) under UV light (365 nm). When the UV light was turned off, the AA-CPDs powders emit a green phosphorescence. The phosphorescence emission spectra have a broad peak from 400 to 600 nm, mainly centered at 476 nm (green line). Under the excitation of WLED, the phosphorescent emission wavelength of AA-CPDs is 464 nm (black line). The phosphorescent lifetime of AA-CPDs were measured at room temperature and shown in Figure 5b. The decay spectrum of AA-CPDs shows their long phosphorescent lifetime, which can be fitted with a three-exponential function. The lifetime components were determined to be 11.96 ms (16.04%), 98.40 ms (40.48%), and 476.08 ms (43.48%), respectively. These results suggest the AA-CPDs possess multiple decay channels. According to the following Equation (2) [56]:(2)$$\tau_{avg}=\frac{\sum B_{i}\tau_{i}{}^{2}}{\sum B_{i}\tau_{i}}$$

The average lifetime was calculated to be 412.03 ms (under excitation at 365 nm). RTP of AA-CPDs was determined and shown in Figure 5c. When the visible/ultraviolet light was removed, the phosphorescence signal of AA-CPDs can still be detected when t = 5 s or 10 s. By comparing the afterglow lifetimes of CDs reported in the literature (Table 1), it can be found that AA-CPDs have a great advantage in this work. Even if ground into powder, RTP performance is still shown, indicating that the crosslink structure is highly resistant to external damage.

### 3.4. RTP Mechanism of AA-CPDs

In order to determine the source of RTP emission, the photophysical properties of the AA-CPDs were thoroughly investigated. Firstly, the AA-CPDs exhibit blue fluorescence (408 nm) in water without afterglow emission. The reason should be due to violent collisions caused by the free movement of molecules [57], causing hydrogen bonds inside AA-CPDs to be destroyed by the solvent effect of water molecules [24] (Figure 6 left). Secondly, the fluorescence spectrum of solid AA-CPDs shows a wide band with a centre wavelength of about 443 nm and a longer tail higher wavelength, which is caused by a phosphorescent phenomenon [1,19]. The AA-CPDs can be closely cross-linked by hydrogen bonds after water is removed, just as solid films are formed [34] (Figure 6 right). The AA-CPDs exhibit non-excitation-related fluorescence spectral characteristics in aqueous solutions; however, solids exhibit emission behaviour associated with excitation wavelengths, indicating that more emission centres are generated in solids that are responsible for the RTP characteristics of excitation dependence. Lastly, FT-IR and XPS analysis show that the AA-CPDs had a polymer/carbon hybrid structure with many groups including carbonyl and amino groups. According to previous reports, the C–N/C=N related groups in AA-CPDs have been confirmed to be the cause of the triplet state correlation emission [22,47,52]. Therefore, suppressing the quenching of triplet excitons generated in the C–N/C=N of CPDs is an effective way to achieve self-protected RTP. Hydrogen bonds effectively lock the emitted material and inhibit its intramolecular motion [13,41,58]. This is through nitrogen doping to achieve the self-protection RTP characteristics of AA-CPDs.

In order to prove that N doping is conducive to the generation of RTP. Therefore, we used oxalic acid instead of ammonium oxalate to prepare contrast CDs without N element, which were named AA-CPDs1. The excitation spectrum of AA-CPDs1 has the only peak located at 205 nm (Figure 7a). The best excitation and emission wavelengths of solid AA-CPDs1 are located at 360 nm and 426 nm, respectively (Figure 7b). Same as AA-CPDs, the solid AA-CPDs1 also has excitation wavelength dependency. The difference is that the fluorescence of AA-CPDs aqueous solution and solid has a red shift compared with that of AA-CPDs1, indicating that the increase of N element leads to the red shift of emission wavelength.

In addition, as can be seen from Figure 8, AA-CPDs1 has only 2 s of afterglow after turning off UV light, which is significantly less than the afterglow time of AA-CPDs. Under the excitation of WLED, AA-CPDs1 does not produce phosphorescence, only ultraviolet light single mode excitation produces phosphorescence. These images illustrate the large difference in RTP performance between the two materials. This result confirms that phosphorescent production is related to nitrogen element. This is consistent with previous reports [34,59] that nitrogen facilitates the n-π* transition, thereby facilitating the forbidden spin transfer from singlet to triplet state excitons through intersystem crossover.

In order to further illustrate that acrylic acid can form long polymer chains during the reaction process, this has been proved in several literatures [60,61]. It has been reported that acrylic acid can form a variety of oligomers, such as dimer, trimer, and tetramer, at high temperatures. Xia [29] et al. reported that carbonized polymer dots (CPDs) include polymer properties, such as rich functional groups, short polymer chains, and incomplete carbonization; meanwhile, a highly crosslinked network structure was formed through dehydration and carbonization [62]. Therefore, we believe that acrylic acid (AA) can form long chain polymers in 180 °C high pressure reactor, and the carbon dots prepared are CPDs. A large number of suspended carboxyl and amino groups on the surface of AA-CPDs can further form a cross-linked structure. The high degree of entanglement and cross-linking structure greatly limits the intramolecular space of AA-CPDs and at the same time shortens the distance between functional groups [50]. The existence of intermolecular interactions inside AA-CPDs can further reduce vibration and rotation, thereby inhibiting non-radiative relaxation [50]. Therefore, the self-matrix structure effectively inhibits the non-radiative transition by blocking the movement of the polymer chain, and can also strongly prevent oxygen quenching, presenting RTP at room temperature. In addition, the carbon dots can find applications in sensing [63,64,65], photodynamic activation on cancer treatment [66,67,68] and imaging [69,70,71], which are the topics for our research.

## 4. Conclusions

In summary, we successfully prepared RTP AA-CPDs using acrylic acid and ammonium oxalate as raw materials. AA-CPDs have the characteristics of producing green RTP under the dual-mode excitation of visible/ultraviolet light. Under environmental conditions, the phosphorescence lifetime of the AA-CPDs solid is 412.03 ms, and the afterglow of visible/ultraviolet light excitation is 5 s and 10 s for naked eye observation, respectively. The carboxyl and amino groups on the surface of AA-CPDs are connected by intermolecular hydrogen bonds. These hydrogen bonds may provide space protection (isolation of oxygen) around the AA-CPDs phosphor, which can stably excite the triplet state. In addition, through comparative experiments, we also verified that N doping promotes the generation of triplet excitons in AA-CPDs. More importantly, the prepared AA-CPDs can be used as a potential new type of smart hidden security material, which can be used in fields such as advanced anti-counterfeiting and the hiding of complex graphics. This research provides a method for simple and rapid preparation of metal-free RTP materials.

## Figures and Tables

**Figure 1 nanomaterials-12-02210-f001:**
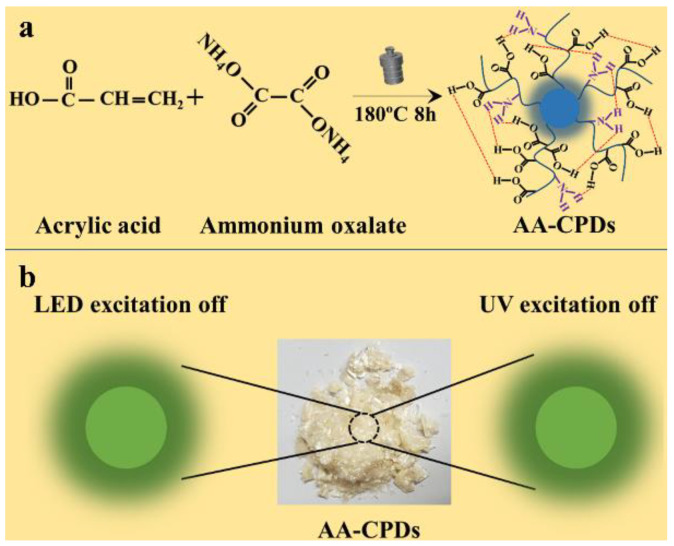
(**a**) Synthetic route of AA-CPDs. (**b**) Schematic illustration of AA-CPDs with WLED and UV excitation (365 nm).

**Figure 2 nanomaterials-12-02210-f002:**
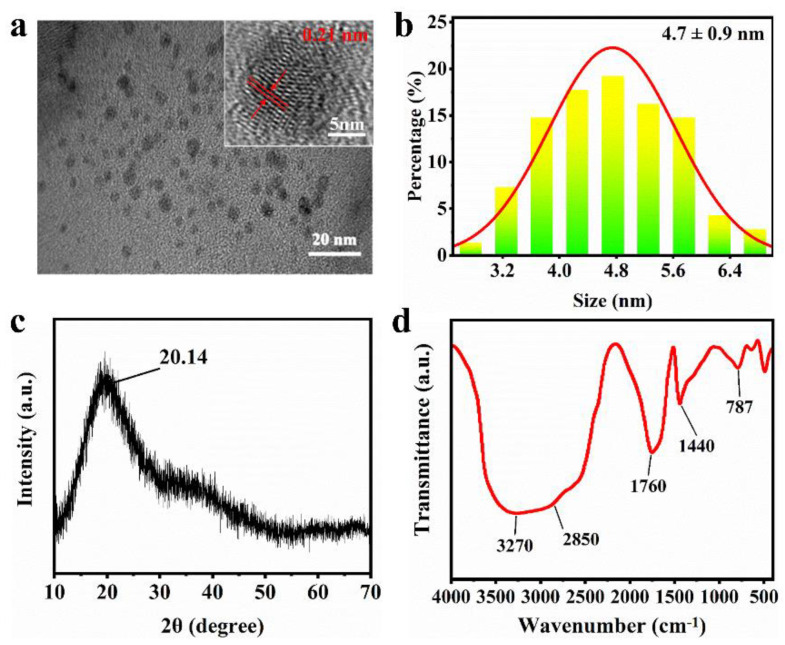
(**a**) HRTEM image, (**b**) Size distribution, (**c**) XRD pattern, and (**d**) FT − IR spectrum of AA-CPDs.

**Figure 3 nanomaterials-12-02210-f003:**
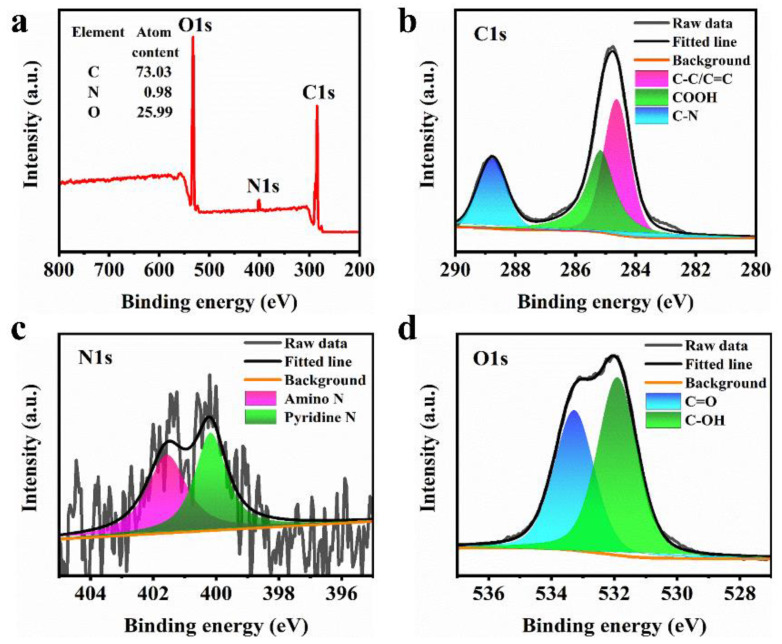
(**a**) XPS spectrum, (**b**–**d**) XPS fitting results for the C1s, N1s, and O1s spectra of AA-CPDs.

**Figure 4 nanomaterials-12-02210-f004:**
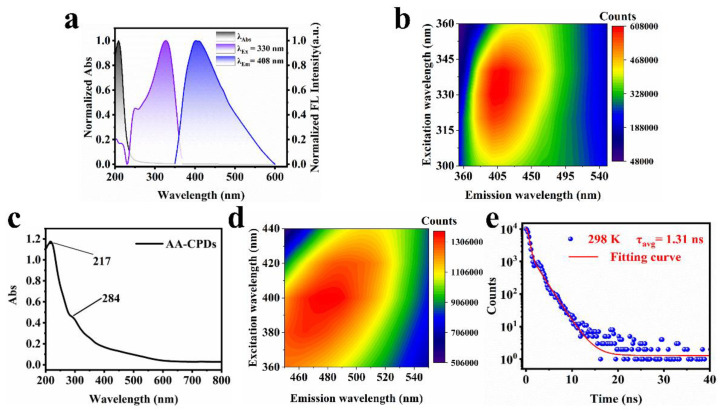
(**a**) The normalized UV–vis absorbance, FL excitation and emission spectra of AA-CPDs in water. (**b**) Fluorescence spectra of AA-CPDs in water. (**c**) UV–vis spectrum of the solid AA-CPDs. (**d**) Fluorescence spectra of the solid AA-CPDs. (**e**) FL decay curve of the AA-CPDs in water.

**Figure 5 nanomaterials-12-02210-f005:**
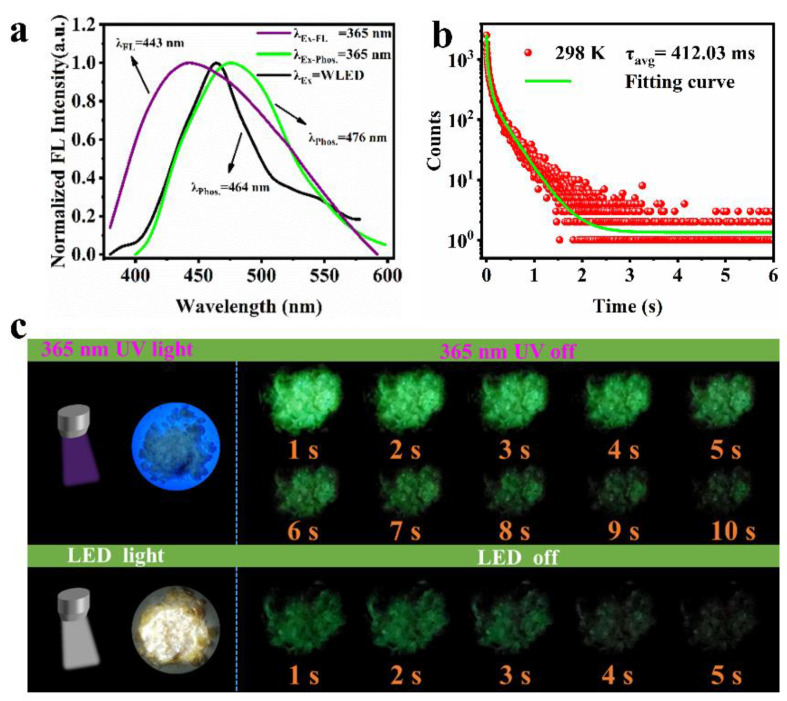
(**a**) FL and afterglow emission spectra of AA-CPDs at 365 nm and WLED. (**b**) Afterglow lifetime of AA-CPDs at 365 nm excitation. (**c**) Images of AA-CPDs powders under UV 365 nm and WLED on and off with time.

**Figure 6 nanomaterials-12-02210-f006:**
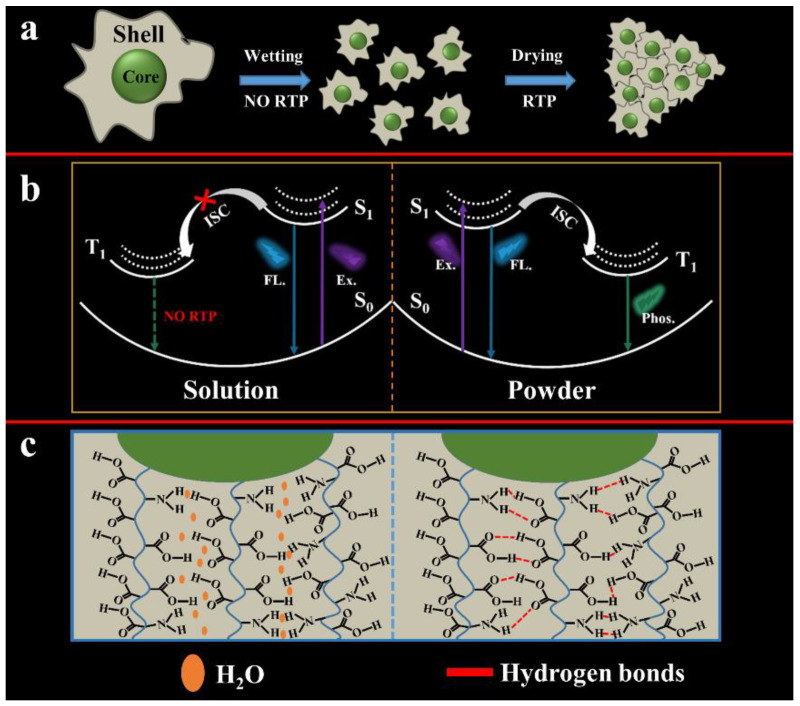
Mechanism of the photophysical processes of AA-CPDs. (**a**) Schematic of the process for achieving RTP; (**b**) The phosphorescence emission process of AA-CPDs; (**c**) Schematic diagram of hydrogen bond network interactions between AA-CPDs.

**Figure 7 nanomaterials-12-02210-f007:**
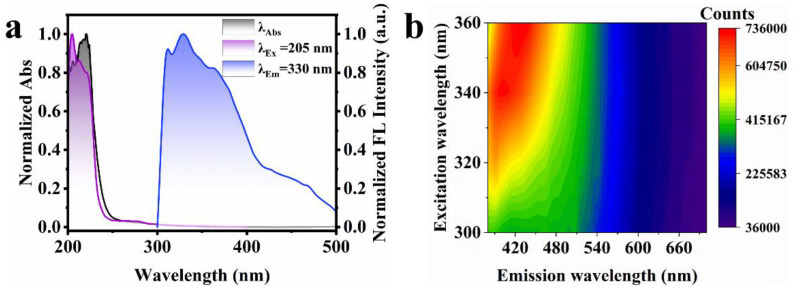
(**a**) The normalized UV absorbance, FL excitation and emission spectra of AA-CPDs1 in water. (**b**) Fluorescence spectra of the solid AA-CPDs1.

**Figure 8 nanomaterials-12-02210-f008:**
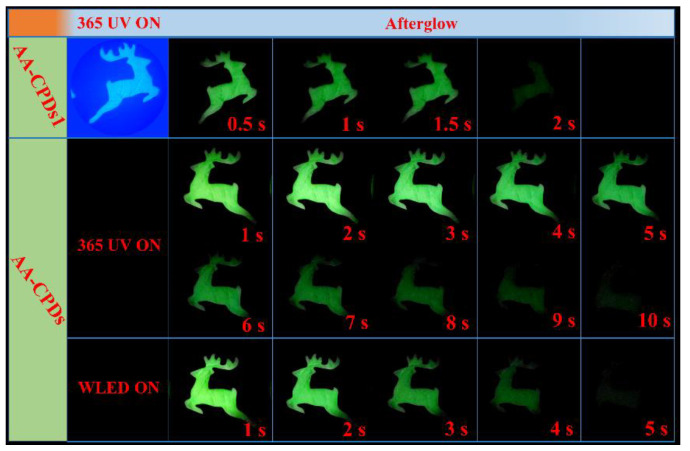
Image of AA-CPDs and AA-CPDs1 powders under UV 365 nm on and off with time.

**Table 1 nanomaterials-12-02210-t001:** A comparison between the decay time to naked eyes and the lifetime of afterglow for the AA-CPDs with those of RTP carbon dots.

RTP Carbon Dots Materials	λ_Em_/nm	Lifetime/ms	Decay Time to Naked Eyes/s	Reference
RTP C-dots	535	750	9	[1]
NCDs	519	459	6	[2]
AA-CDs	585	240.8	5	[21]
F-CDs	540	1390	10	[22]
P-CDs	525	1140	9	[25]
PCDs Ⅰ-1	494	658.11	7	[37]
PCDs Ⅰ-2	--	379.22	1.4	[37]
PCDs Ⅰ-3	--	188.58	0.2	[37]
NCDs	515	747	2	[24]
AN-CPDs-150	485	373.5	--	[31]
AN-CPDs-180	490	436.1	--	[31]
AN-CPDs-200	494	466.5	--	[31]
AN-CPDs-230	500	257.8	--	[31]
AN-CPDs-250	518	174.8	--	[31]
AN-CPDs-280	532	117.2	--	[31]
AN-CPDs-300	558	61.4	--	[31]
AA-CPDs	476	412.03	10	This work

## Data Availability

Not applicable.

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
