# Peer review of "Self-Matrix N-Doped Room Temperature Phosphorescent Carbon Dots Triggered by Visible and Ultraviolet Light Dual Modes"

_nanomaterials, 2022, doi:10.3390/nano12132210_

Round 1

Reviewer 1 Report

The work is interesting but significant revision is required before publication can be considered.

1.     Fitting curves for the PL decay (“triadic exponential decay") need to be included. My experience is that these systems are better characterized by stretched exponential decays representative of a broad range of decay rates. A log(t) axis in Fig. 4d would also be more informative than linear, since it appears there is not a single exponential decay. I am also curious about the baseline in Fig. 4d and how this impacts the fitting. Panels (b) and (c) should be presented in the same manner. It isn’t clear what (b) means as shown (two plots).

2.     I have the same criticism for Fig. 5b. Where is the fitting curve, and why only three terms and not infinite (stretched exponential decay). If there are only three channels, this will be obvious on a log-log scale. I suspect it is more complex than this, but it currently reads as if the authors could not get a decent fit with one (obviously since the plot in Fig. 5b is not linear), so they added terms until there was convergence. However, a true multi-exponential (“stretched”) decay appears like it would provide a better description. What is the physical basis for only including three terms? There are likely many more than three states from which the decay originates.

3.     Again, the double sheet plot of Fig. 7b is unneeded and confusing.

4.     The color-schemes of Fig. 4a and 7a should match. The absorption spectrum should be gray for each, etc. Also, Fig. 7a and b don’t appear to match. The peak in the “excitation spectrum” looks like it should be closer to 250 nm.

5.     The paragraph above Fig. 8 is too long and contains multiple speculative points with meaning that is very unclear. Examples: “It has the unique characteristics of high oxygen content and excellent water solubility, which is due to the polymer structure and special photoluminescence mechanism” “The self-matrix structure effectively inhibits the non-radiative transition by blocking the movement of the polymer chain, and can also strongly prevent oxygen quenching, presenting RTP at room temperature” – How do trends toward more glassy polymer behavior inhibit non-radiative transitions? What is the precise mechanism?

6.     Section 3.5 is a stretch and should probably not be included in the paper. The application potential is obscure and statements like “Dual mode anti-counterfeiting can be achieved in ultraviolet/visible light, increasing the accuracy and difficulty of security” make no sense.

Author Response

The work is interesting but significant revision is required before publication can be considered.

  1. Fitting curves for the PL decay (“triadic exponential decay") need to be included. My experience is that these systems are better characterized by stretched exponential decays representative of a broad range of decay rates. A log(t) axis in Fig. 4d would also be more informative than linear, since it appears there is not a single exponential decay. I am also curious about the baseline in Fig. 4d and how this impacts the fitting. Panels (b) and (c) should be presented in the same manner. It isn’t clear what (b) means as shown (two plots).

Response: Thanks! we have added the fitting curve in Fig. 4d. The fluorescence lifetime curve is fitted under the nonlinear curve, which conforms to the three exponential Lagrange function, and the ordinate is actually log(10). The fitting curve has converged. This diagram has no baseline. 

Figures 4b and 4c have been modified in the same manner according to your suggestion.

  1. I have the same criticism for Fig. 5b. Where is the fitting curve, and why only three terms and not infinite (stretched exponential decay). If there are only three channels, this will be obvious on a log-log scale. I suspect it is more complex than this, but it currently reads as if the authors could not get a decent fit with one (obviously since the plot in Fig. 5b is not linear), so they added terms until there was convergence. However, a true multi-exponential (“stretched”) decay appears like it would provide a better description. What is the physical basis for only including three terms? There are likely many more than three states from which the decay originates.

Response: Thanks! we have added the fitting curve in Fig. 5b. The number of channels may be more than 3, but the correlation coefficient of fitting with the three-exponential Lagrange function has reached 0.99, so it can be considered that the fitting curve has converged.

  1. Again, the double sheet plot of Fig. 7b is unneeded and confusing.

 Response: Thanks! we have deleted Fig. 7b in the manuscript according to your suggestion.

  1. The color-schemes of Fig. 4a and 7a should match. The absorption spectrum should be gray for each, etc. Also, Fig. 7a and b don’t appear to match. The peak in the “excitation spectrum” looks like it should be closer to 250 nm.

 Response: Thanks! We have changed the color of Figure 7a to match that of Figure 4a. From the excitation spectrum in Fig. 7a, it can be seen that the excitation wavelength is 205nm.

  1. The paragraph above Fig. 8 is too long and contains multiple speculative points with meaning that is very unclear. Examples: “It has the unique characteristics of high oxygen content and excellent water solubility, which is due to the polymer structure and special photoluminescence mechanism” “The self-matrix structure effectively inhibits the non-radiative transition by blocking the movement of the polymer chain, and can also strongly prevent oxygen quenching, presenting RTP at room temperature” – How do trends toward more glassy polymer behavior inhibit non-radiative transitions? What is the precise mechanism?

  Response: Thanks! We have rewritten the paragraph above Figure 8. The mechanism is simple and clear.

  1. Section 3.5 is a stretch and should probably not be included in the paper. The application potential is obscure and statements like “Dual mode anti-counterfeiting can be achieved in ultraviolet/visible light, increasing the accuracy and difficulty of security” make no sense.

Response: Thanks! we have deleted Section 3.5 in the manuscript according to your suggestion.

Reviewer 2 Report

Authors have synthesized and characterized a nanomaterial that emits phosphorescence under visible/ultraviolet light. The authors have described all the steps carried out in the synthesis and characterization of Carbon Dots, and have tested their phosphorescent properties for use in fraud detection. In my opinion, this paper could be published after minor revisions, because there are some items that the authors should explain:

·         There are other works on AA-CDs and CPDsbibliographies such as the cited in: DOI:10.3390/nano10030464 and https://doi.org/10.1002/advs.201901316

 Information about these types of research works should be included in the Introduction section in order to understand that in your research work you are combining AA- and CPDs.

·         In page 4, line 136 the authors cite:

“To understand the origin of phosphorescence, the morphology and particle size of 136 the AA-CPDs were investigated”

After that authors describe the results of analysing CPDs by HRTEM, XRD, FTIR, etc. but they do not clarify the relation between RTP and characteristics.

·         In line 193 authors wrote:

·         The excitation and emission wavelengths of solid-state fluorescence are located at 400 nm and 478 nm, respectively. When the excitation wavelengths change from 360 nm to 440 nm, the maximum emission wavelengths of solid AA-CPDs change from 440 nm to 506 nm.”

Please clarify the text:

1.      Location of maximum: For excitation 400nm emission is 478 nm

2.      For excitation 360 nm emission is 440 nm

3.      For excitation 440 nm emission is 506 nm

It is not clear the reason for moving the excitation and emission wavelengths of the solid-state. What do you want to get doing these changes? Please explain this in the text

·         In line 206, the decay profiles are studied at 365 nm. Why there is not uniformity in wavelength selection? Explain reasons in the text.

·         Figure 7 it is not clearly explained. Why 7b and 7c have been shown as different Figures? Both show emission, excitation and counts of signals.

·         Conclusions line 331:

“AA-CPDs have good water solubility and have the characteristics of producing green RTP under the dual-mode excitation of visible/ultraviolet light.”

Line 240 section 3.4:

“the AA-CPDs exhibit blue fluorescence (408 nm) in water without afterglow emission

Conclusions and section  3.4 are contradictory responses, please clarify and explain these results properly in the  text and conclusions.

Author Response

Authors have synthesized and characterized a nanomaterial that emits phosphorescence under visible/ultraviolet light. The authors have described all the steps carried out in the synthesis and characterization of Carbon Dots, and have tested their phosphorescent properties for use in fraud detection. In my opinion, this paper could be published after minor revisions, because there are some items that the authors should explain:

  There are other works on AA-CDs and CPDs bibliographies such as the cited in: DOI:10.3390/nano10030464 and https://doi.org/10.1002/advs.201901316

 Information about these types of research works should be included in the Introduction section in order to understand that in your research work you are combining AA- and CPDs.

Response: Thanks! For the two references, see references 21and 29 in the manuscript.

In page 4, line 136 the authors cite

“To understand the origin of phosphorescence, the morphology and particle size of the AA-CPDs were investigated” After that authors describe the results of analysing CPDs by HRTEM, XRD, FTIR, etc. but they do not clarify the relation between RTP and characteristics.

Response: Thank you! We've rewritten that sentence.

  • In line 193 authors wrote:
  • The excitation and emission wavelengths of solid-state fluorescence are located at 400 nm and 478 nm, respectively. When the excitation wavelengths change from 360 nm to 440 nm, the maximum emission wavelengths of solid AA-CPDs change from 440 nm to 506 nm.”

 Please clarify the text:

  1. Location of maximum: For excitation 400nm emission is 478 nm
  2. For excitation 360 nm emission is 440 nm
  3. For excitation 440 nm emission is 506 nm

It is not clear the reason for moving the excitation and emission wavelengths of the solid-state. What do you want to get doing these changes? Please explain this in the text.

Response: Thank you! These data show that the fluorescence spectrum of solid AA-CPDs is wavelength-dependent, they have different emission wavelengths for different excitation wavelengths. These data only reflect the inherent properties of the solid AA-CPDs themselves and this is a common phenomenon of carbon dots.

In line 206, the decay profiles are studied at 365 nm. Why there is no uniformity in wavelength selection? Explain the reasons in the text.

Response: Because the phosphorescence lifetime is measured at 365 nm, we want to study how the fluorescence life of AA-CPDs is at 365 nm as 365 nm is a common wavelength in UV lams that is often used for phosphorescence activation. Therefore, the fluorescence lifetime of  AA-CPDs is not measured at this wavelength.

  • Figure 7 is not clearly explained. Why 7b and 7c have been shown as different Figures? Both show emission, excitation, and counts of signals.

Response: Thanks! we have deleted Fig. 7b in the manuscript according to another reviewer's suggestion. Figure 7 shows the fluorescence spectrum of carbon dots without N element. This is a reference carbon dots (named AA-CPDs1) to prove that N-doping is beneficial to phosphorescent production. The absorption and fluorescence spectra of the reference carbon dots are different from the AA-CPDs we studied.

 Conclusions line 331:

“AA-CPDs have good water solubility and have the characteristics of producing green RTP under the dual-mode excitation of visible/ultraviolet light.”

Line 240 section 3.4:

“the AA-CPDs exhibit blue fluorescence (408 nm) in water without afterglow emission

Conclusions and section  3.4 are contradictory responses, please clarify and explain these results properly in the text and conclusions.

Response: Thanks! We have deleted “AA-CPDs have good water solubility”.

We just want to tell that carbon dots are very water-soluble,  we did not mean that they have phosphorescence in aqueous solution.

Reviewer 3 Report

In this manuscript, the author reports, ‘Self-matrix N-doped room temperature phosphorescent carbon dots triggered by visible and ultraviolet light dual modes’. The authors should address the following questions before getting a possible publication.

 Recommendation: Major revisions needed as noted.

1.      The novelty of the present work should be discussed in the Introduction section.

2.      The author should write the purpose for each test in one/two sentences (in brief) before explaining the results of the characterization techniques. Therefore, the logic and organization of this part will be enhanced.

3.  The authors are encouraged to provide UV-vis absorption spectrum of the AACPDs.

4.      What about the photostability of the AACPDs?

5.   The formatting and grammatical errors in the article need to be checked carefully.

6.      Did the author calculate quantum yield of the AACPDs?

7.       The authors have cited relevant references in the Introduction section; however the manuscript needs to be highlighted with some recent reports to further broaden the impact, related literatures: Sensors and Actuators B: Chemical, 304, 127344; Journal of Membrane Science 639 (2021): 11975; Langmuir, 37(11), 3508-3520; ACS Applied Materials & Interfaces, 7(49), 27324-27330; Nanomaterials, 11(2), 369

Author Response

Reviewer-3

In this manuscript, the author reports, ‘Self-matrix N-doped room temperature phosphorescent carbon dots triggered by visible and ultraviolet light dual modes’. The authors should address the following questions before getting a possible publication.

 Recommendation: Major revisions needed as noted.

1. The novelty of the present work should be discussed in the Introduction section.
Response: Thanks! we have made modifications in the manuscript according to your suggestion.

2. The author should write the purpose for each test in one/two sentences (in brief) before explaining the results of the characterization techniques. Therefore, the logic and organization of this part will be enhanced.
Response: Thanks! we have made modifications in the manuscript according to your suggestion.

3. The authors are encouraged to provide UV-vis absorption spectrum of the AACPDs.
Response: Thanks! we have made UV-vis absorption spectrum of the AA-CPDs (Fig.4c).

4. What about the photostability of the AACPDs?
Response: The AA-CPDs have good optical stability in sealed condition. After repeating irradiation with 365nm UV lamp, the decay time of naked eye afterglow is basically unchanged.

5. The formatting and grammatical errors in the article need to be checked carefully.
Response: Thank! We have checked carefully the English. A few errors have been modified.

6. Did the author calculate quantum yield of the AACPDs?
Response: Yes. The quantum yield is given on page 6, line 213.

7. The authors have cited relevant references in the Introduction section; however the manuscript needs to be highlighted with some recent reports to further broaden the impact, related literatures: Sensors and Actuators B: Chemical, 304, 127344; Journal of Membrane Science 639 (2021): 119754; Langmuir, 37(11), 3508-3520; ACS Applied Materials & Interfaces, 7(49), 27324-27330; Nanomaterials, 11(2), 369
Response: The references have been written in the manuscript according to your suggestion. See references  3, 4, 37, 42, 44.

Round 2

Reviewer 1 Report

I find the revisions made by the authors to be sufficient for acceptance in Nanomaterials

Reviewer 3 Report

The authors have addressed all the questions raised before. Therefore the manuscript can be accepted in the present form